

# Quantitative estimation of wastewater quality parameters by hyperspectral band screening using GC, VIP and SPA

Zheng Xing[1,2], Junying Chen[1,2], Xiao Zhao[1,2], Yu Li[2], Xianwen Li[2], Zhitao Zhang[1,2], Congcong Lao[2] and Haifeng Wang[2]

[1] Key Laboratory of Agricultural Soil and Water Engineering in Arid and Semiarid Areas, Ministry of Education, Northwest A&F University, Yangling, Shaanxi, China
[2] College of Water Resources and Architectural Engineering, Northwest A&F University, Yangling, Shaanxi, China

Corresponding author
Junying Chen, cjyrose@126.com

## ABSTRACT

Water pollution has been hindering the world's sustainable development. The accurate inversion of water quality parameters in sewage with visible-near infrared spectroscopy can improve the effectiveness and rational utilization and management of water resources. However, the accuracy of spectral models of water quality parameters is usually prone to noise information and high dimensionality of spectral data. This study aimed to enhance the model accuracy through optimizing the spectral models based on the sensitive spectral intervals of different water quality parameters. To this end, six kinds of sewage water taken from a biological sewage treatment plant went through laboratory physical and chemical tests. In total, 87 samples of sewage water were obtained by adding different amount of pure water to them. The raw reflectance ($R_{raw}$) of the samples were collected with analytical spectral devices. The $R_{raw-SNV}$ were obtained from the $R_{raw}$ processed with the standard normal variable. Then, the sensitive spectral intervals of each of the six water quality parameters, namely, chemical oxygen demand (COD), biological oxygen demand (BOD), $NH_3$-N, the total dissolved substances (TDS), total hardness (TH) and total alkalinity (TA), were selected using three different methods: gray correlation (GC), variable importance in projection (VIP) and set pair analysis (SPA). Finally, the performance of both extreme learning machine (ELM) and partial least squares regression (PLSR) was investigated based on the sensitive spectral intervals. The results demonstrated that the model accuracy based on the sensitive spectral ranges screened through different methods appeared different. The GC method had better performance in reducing the redundancy and the VIP method was better in information preservation. The SPA method could make the optimal trade-offs between information preservation and redundancy reduction and it could retain maximal spectral band intervals with good response to the inversion parameters. The accuracy of the models based on varied sensitive spectral ranges selected by the three analysis methods was different: the GC was the highest, the SPA came next and the VIP was the lowest. On the whole, PLSR and ELM both achieved satisfying model accuracy, but the prediction accuracy of the latter was higher than the former. Great differences existed among the optimal inversion accuracy of different water quality parameters: COD, BOD and TN were very high; TA relatively high; and TDS and TH relatively low. These findings can

provide a new way to optimize the spectral model of wastewater biochemical parameters and thus improve its prediction precision.

## INTRODUCTION

Water pollution, one of the most important causes for the shortage of the utilizable water resources, has seriously threatened the sustainable development of human society. According to the statistics of the 4th World Water Forum UN World Water Development Report, about 420 billion cubic meters of sewage is poured into rivers and lakes every year, polluting 550 million cubic meters of fresh water, which is equivalent to over 14% of the total amount of global river. Water quality is affected by the synergy of multiple pollutants (*Brönmark & Hansson, 2002*). The excessive presence of these pollutant particles would make the water quality parameters go beyond the current standards, so the water becomes too contaminated to use. Therefore, accurately and efficiently acquiring the water quality parameters of these pollutants can improve the pertinence and effectiveness of water pollution monitoring.

The traditional procedure of quantitatively estimating the water pollutant parameters mainly includes such steps as water sampling at fixed points, physical and chemical examinations in laboratory, and comprehensive statistical analysis. Such a method is not only time-consuming and laborious, but also limited in observation points and representation (*Esterby, 1996*; *Shafique et al., 2003*). There are more than 117 million lakes on the earth, only a very small number of which are under regular and continuous monitoring (*Verpoorter et al., 2014*). Due to the limitations of the traditional method, it is impossible to achieve dynamic monitoring of water quality in large areas.

Different from the conventional methods, remote sensing technology can acquire large-area water spectral information quickly, continuously and inexpensively and obtain by different methods the multiple components information of water from remote sensing images (*Bukata & Bukata, 2005*; *Arabi et al., 2016*; *Campanelli et al., 2017*; *Deng et al., 2017*). However, due to the influence of spatial and temporal variation and long transmission distance, about 90–98% of the signals obtained by the remote sensing are from the surface of the water and the atmosphere, and the remaining 2–10% of the signals are from the water components. This has led to the complexity of the optical characteristics of water, which has made it rather difficult to identify the information of water pollutants, resulting in the uncertainty in the extraction results of water pollutants (*Gitelson & Kondratyev, 1991*). In the water, some of the main pollutant parameters (chemical oxygen demand (COD), biological oxygen demand (BOD)) change the radiation of light through their own absorption and scattering characteristics of light, and thus have different characteristic absorption spectra, which strongly correlates the pollutant content and spectral reflectance (*Duan et al., 2006*; *Cao et al., 2015*; *Wu, Du & Yan, 2011*).

In recent years, hyper-spectral techniques are still being widely used for various monitoring of natural resources. Nowadays, the application of hyper-spectral techniques can facilitate the elimination of the external interference, obtain fine spectral information, and ensure the accuracy of spectral inversion. Therefore, more researchers have deeply studied the application of the hyperspectral technology in obtaining the visible and near-infrared (VIS-NIR) band in order to estimate the content of some pollutants (*Palmer, Kutser & Hunter, 2015*; *Han & Rundquist, 1997*).

At present, the inversion of water quality parameters is mainly aimed at the physiological parameters of the water, such as transparency and nutrients (*Zhang, Giardino & Li, 2017*), but rarely at such biochemical parameters of water as its total hardness (TH), total dissolved substances (TDS), BOD (*Yang, Yan & Lin, 2004*), COD (*Rojas, 2009*), total alkalinity (TA) and ammonia nitrogen ($NH_3$-N) (*Lerch et al., 2015*). These water quality parameters reflect the water pollution degree, which is of great significance for the purposeful and effective treatment of sewage. Among the spectral inversion methods of water quality parameters, the empirical statistical method remains the mainstream. The partial least squares regression (PLSR) (*Wold, 1966*), as a typical linear regression method, is of wide use in model construction for its advantage of using all available bands without multi-collinearity problem. With the gradual promotion of machine learning algorithms, such advanced semi-empirical machine learning algorithms (*Keller et al., 2018*) as artificial neural networks (*Isiyaka et al., 2018*; *Bansal & Ganesan, 2019*), support vector machine, extreme learning machine (ELM) have also been gradually applied in the retrieval of water quality parameters, which have greatly improved the model inversion accuracy. Compared with traditional neural networks, ELM calculates faster with the learning accuracy guaranteed. However, in the process of spectral data modeling, there are two major problems in data processing caused by the large amount of data information of hyperspectrum (*Zhang, Maoguo & Yongqiang, 2018*). One problem is that between the adjacent frequency bands usually has a high correlation, which brings huge redundant information. Redundant information is of no use to the processing of hyperspectral data, which will be a great waste of computation and storage. Another problem is that the detailed information of spectrum will increase the dimension of hyperspectral data, which may lead to the "Hughes phenomenon." Hughes phenomenon means the increase of data dimension will decrease the classification accuracy when the number of training samples is limited (*Hughes, 1968*). Therefore, in current researches, scholars are trying to adopt some methods to screen the characteristic spectrum, to remove redundant information, and to improve the stability and predictability of the model (*Feng et al., 2018*). The variable importance in projection (VIP) algorithm has successfully distinguished the hyperspectral band subsets of forest species by identifying the importance of the independent variables (*Peerbhay, Mutanga & Ismail, 2013*); the gray correlation (GC) method, using the gray values of the independent variables, has achieved the screening of spectral intervals of soil ion characteristics (*Wang et al., 2019*). So far these two methods have rarely been applied in the screening of the characteristic spectrum of water pollution parameters. Meanwhile, these band screening methods have different preferences and significant differences in the performance of different modeling methods,

which will lead to the uncertainty in the model accuracy (*Lerch et al., 2015*). Iterative progressive elimination combined with PLSR has a good performance in the inversion of chlorophyll A and total suspended solids (*Wang et al., 2018a*). VIP combined with PLSR and SVR has significantly improved the inversion effect of characteristic spectra of water-soluble ions in soil, while GC combined with PLSR and SVR showed relatively poor inversion effect (*Wang et al., 2019*). However, GC combined with ELM has a good effect on the prediction and inversion of total nitrogen content in soil (*Zhou et al., 2017*). Moreover, in order for the better universality and stability of band selection method, the set pair analysis (SPA) theory was used to establish models which can identify the overall degree as well as the important indicators and subsystems in the influencing factors (*Zhang et al., 2019*). SPA theory, for its remarkable capability of dealing with uncertain problems, has been widely applied in many fields (*Wang et al., 2017a*; *Li et al., 2016*), but not yet applied in band selection. This research applied the band selection methods of GC, VIP and SPA to select from hyperspectral data the feature subsets, which were used to establish PLSR and ELM models to estimate such water quality parameters as COD, BOD, $NH_3$-N, TDS, TH and TA, and to compare and analyze the inversion effects of different pollutant indicators.

In particular, the study aims to: (1) determine the optimal spectral interval corresponding to the six water pollution indicators, build the optimal inversion models of six water pollution parameters, and achieve quantitative estimation of water pollution parameters through hyperspectrum; (2) verify the SPA method in weighing the different wavelength screening methods and provide an approach to band selection by applying the set optimization idea; (3) determine the corresponding optimal models for different water quality parameters by comparing the performance of GC, VIP and SPA models.

# MATERIALS AND METHODS

## Sample preparation and chemical analysis

The water samples were taken from different spots (water inlet, anoxic tank, aerobic tank, sedimentation tank and water outlet) under different treatment methods at a domestic sewage treatment plant in China. The domestic sewage treatment plant gave field permit approval to us (NO. 51409221, 51979234). According to our pre-analysis (Table 1) (*CESP, 2002*; *Kotti et al., 2018*), the water samples collected at the inlet were abundantly rich in COD, BOD and NH3-N (COD: 425 mg/L, BOD: 101.15 mg/L, NH3-N: 34.853 mg/ L), whilst the water samples collected in other treatment ponds were relatively low in COD of 20–140 mg/L, BOD of 3.1–13 mg/L and NH3-N of 0–1.723 mg/L. Such large variations suited well to represent the range of water quality collected from different locations investigated in this study, but also leaving non-negligible deficiency in representing water quality of moderate concentrations. Therefore, to make the hyperspectral model more robust and responsive to minor changes of water quality at other potential locations, we further diluted the water samples into subsamples to form a full range of distribution within the minimum and maximum values detected in this study. To be specific, following the minimum and maximum COD measured from the inlet and other treatment ponds,

**Table 1 Main water quality parameters.**

| Parameters | Water inlet | Anoxic tank | Aerobic tank | Sedimentation tank | Outlet of water | Experiment methods |
|---|---|---|---|---|---|---|
| $NH_3$-N (mg/L) | 34.853 | 1.723 | 1.499 | 0 | 0 | According to Nessler's reagent spectrophotometer with the amount of visible light spectrophotometer 722 N for determining $NH_3$-N |
| Total alkalinity (mg/L) | 251.70 | 147.02 | 148.20 | 101.15 | 103.50 | According to acid base indicator titration method |
| Total hardness (mmol/L) | 1.09 | 1.13 | 1.13 | 1.17 | 1.11 | According to the EDTA titration method (GB11914-1989) |
| Total dissolved substance (mg/L) | 351 | 323 | 317 | 344 | 343 | According to Gravimetric method (GB T5750.5-2006) |
| COD (mg/L) | 425 | 140 | 134 | 23 | 20 | According to the dichromate method (GB11914-1989) using a standard COD digestion apparatus (K-100) to determine COD |
| BOD (mg/L) | 86 | 8.5 | 13 | 3.1 | 6.2 | According to the dilution and inoculation method (HJ505-2009) with a constant temperature incubator (HWS-150 type) for determining the content of BOD5 |

the water samples rich in COD were diluted by adding different amount of pure water, forming different COD concentrations varying at unit of five mg/L. In total, 87 water samples were obtained to establish the hyperspectral model and strengthen its applicability. In a real application of hyperspectral model, the water quality of targeted samples often covers a variety of concentrations, and thus can be measured and applied directly without any dilution. The laboratory was kept in a constant temperature during the experiments. The main water quality parameters included $NH_3$-N, TA, TH, TDS, COD and BOD.

## Acquisition and pretreatment of spectral data

The wastewater samples were put into black cylindrical cups with a depth of five cm and a diameter of 10 cm for spectral data collection in the laboratory. The hyperspectral data for wastewater samples were measured using an ASD (Analytical Spectral Devices, Inc., Boulder, CO, USA) FieldSpec® 3 spectrometer with spectral range from 350 to 2,500 nm. The instrument was equipped with one sensor with 1.4 nm spectral resolution for the range of 350–1,000 nm and the other sensor two nm, 1,000–2,500 nm. The spectral data were collected in a dark room by exposing the water sample surfaces to a halogen lamp of 50 W above with a 30° of incident angle and 50 cm in distance and a fiber-optics probe 5°, 15 cm, so as to minimize the effects of external factors (*Wang & Zhao, 2000*). Before each measurement, we had fully preheated the spectrometer and light source and checked the spectrometer with a standardized white panel of 99% reflectance in order to reduce measurement errors. Each water sample was measured twice vertically, and at each of which the spectral data were gathered 10 times. Altogether there were 20 spectrum curves for each sample (*Hong et al., 2018*). From the 20 curves the raw spectral reflectance ($R_{raw}$), namely, arithmetic mean, were obtained by using ViewSpecPro software (6.0 version). The fluctuation of $R_{raw}$ would influence the accuracy of subsequent modeling due to the disturbance of the random error, instrument noise and external environment in

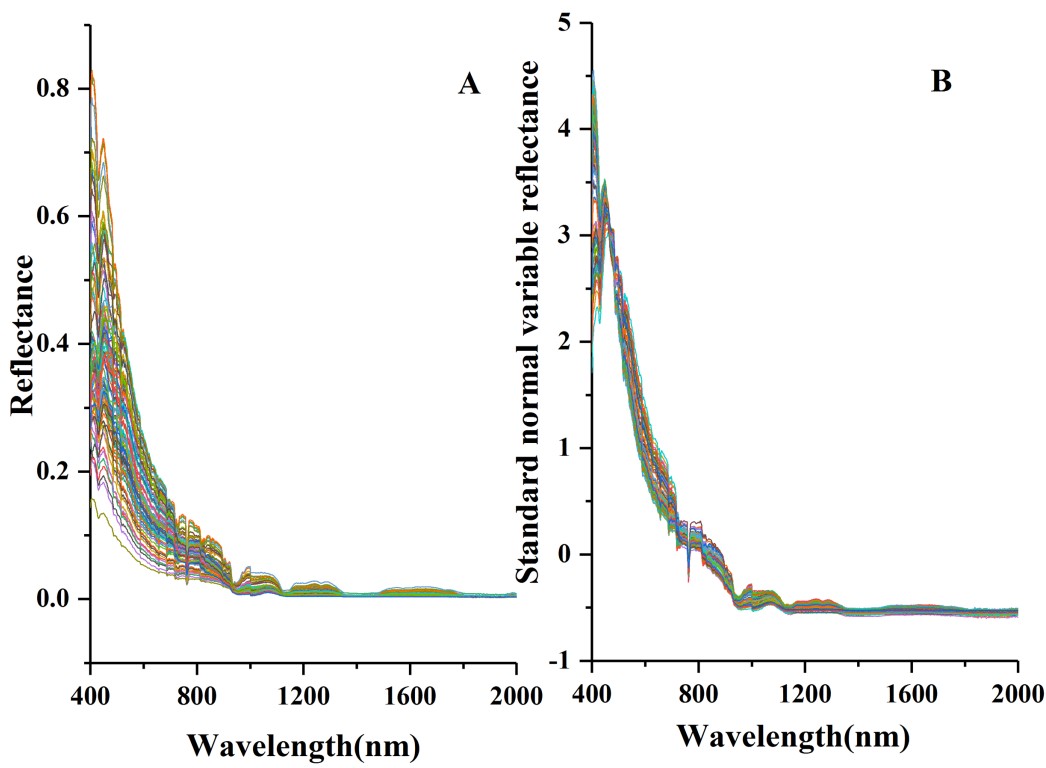

**Figure 1 Spectral data curve.** (A) Reflectance spectral curves. (B) Standard normal variable reflectance curves.

spectral data collection. In general, the external noise can be eliminated to some degree with such effective pretreatments as resampling, smoothing and transformation, which can improve the spectral characteristics (*Ding et al., 2018*). Therefore, two steps were adopted to pretreat the $R_{raw}$: (1) removing the marginal wavelengths (2,401–2,500 nm and 350–399 nm) of higher noise in each water sample, then smoothing the remaining spectrum data through filter method (polynomial order = 2; window size = 5) of Savitzky-Golay (*Savitzky & Golay, 1964*) by Origin Pro software (2017SR2 version); and (2) obtaining the precise $R_{raw-SNV}$ using the standard normal variable (SNV) to remove the effects of baseline shift and surface scattering on the spectral data (*Xiao, Yi & Hao, 2016*). The spectral curves of $R_{raw}$ and $R_{raw-SNV}$ are shown in Figs. 1A and 1B. As can be seen, the spectral curve in Fig. 1B is much smoother than that in Fig. 1A.

## Wavelength selection methods

### Gray correlation

The GC method, one of the gray system theories, seeks the primary and secondary relations and analyzes the different effects of all the factors in a system (*Ju-Long, 1982*; *Liu, Yang & Wu, 2015*). The gray correlation degree (GCD), varying from one moment and object to the other, refers to the measurement of the factor correlation between two systems. In the process of system development, higher consistency of change trend of the two factors reflects higher correlation degree and vice versa (*Zhu et al., 2017*). The main

calculation steps are: first, determining signature and factor sequences with the former marked as $x_0(t)$ which is a collection of $m$ data—$x_0(t) = \{x_0(1), x_0(2),\ldots, x_0(m)\}$ and the latter marked as $x_i(t)$ which contains $n$ sub-sequences with each as a collection of $m$ data—$x_i(t) = \{x_i(1), x_i(2),\ldots, x_i(m)\}$; and second, processing the data sequences dimensionlessly using averaging operators. $X_i = \{x_i(1), x_i(2),\ldots, x_i(m)\}$ represents the behavior sequence of factor $x_i$ and $D$ the averaging sequence operator, and thus $X_iD = \{x_i(1)d, x_i(2)d,\ldots, x_i(m)d\}$. So the mean data value is calculated as

$$x_i(k)d = \frac{x_i(k)}{\overline{X_i}} \quad \overline{X_i} = \frac{1}{n}\sum_{k=1}^{n} x_i(k) \quad k = 1, 2, .., n \tag{1}$$

$$\xi_{0i}(t) = \frac{\Delta_{\min} + \rho\Delta_{\max}}{\Delta_{0i(t)} + \rho\Delta_{\max}} \tag{2}$$

where $\Delta_{\max}$ is the maximum of $|x_0(t) - x_i(t)|$; $\Delta_{\min}$ is the minimum of $|x_0(t) - x_i(t)|$; $|x_0(t) - x_i(t)|$ is the value at time $t$; $\rho$ is the coefficient of discrimination. Therefore, GCD between the signature and factor sequences is calculated as

$$\gamma_{0i} = \frac{1}{n}\sum_{t=1}^{n} \xi3_{0i}(t) \tag{3}$$

From the interaction between the signature and factor sequences, the factors' primary and secondary influences can be predicted (*Wang et al., 2018b*). In this paper, the higher the GCD of a certain band is, the more sensitive the band is to the water quality parameter, and vice versa.

## Variable importance projection

The VIP is a variable screening analysis method based on PLSR model. The VIP value of a spectral variable reflects the importance of this variable in the prediction of the substance to be measured (*Oussama et al., 2012*; *De Almeida et al., 2013*). The VIP value reflects the explanatory power of independent variable over dependent variable, and represents the importance of independent variable to model fitting (*Chavana-Bryant et al., 2019*). If the explanatory ability of each variable to $Y$ is the same, then the VIP value of all independent variables is 1. If the VIP value of an independent variable is less than 1, it means that the variable makes little contribution to the model and has a very low ability to explain the dependent variable, so it can be eliminated. In the PLSR model, $VIP_i$ to an independent variable $X_i$ is defined as:

$$VIP_i = \sqrt{\frac{p * \sum_{h=1}^{m} Rd(Y; t_h)W_{hj}^2}{Rd(Y; t_1, t_2, \cdots, t_m)}} \tag{4}$$

where $p$ is the dimension of independent variable; $t_h$ is the component $h$; $m$ is the number of components selected in the model; $W_{hj}$ is the component of $X_j$ corresponding to axis $W_h$; $Rd(Y; t_h)$ represents the variation of $Y$ explained by the component $t_h$; and $Rd(Y; t_1, t_2, \cdots t_m)$ represents the cumulative variation accuracy of $Y$ explained by $t_1 - t_m$.

In PLSR model, the explanatory power of $x_j$ over $Y$ is passed through $t_h$, so if $t_h$ bears a strong explanatory power over Y, and meanwhile $x_j$ plays a very important role in the process of $t_h$ construction, $t_h$ can be reasonably regarded as very important variable in explaining $Y$. In this way of wavelengths selection, the wavelengths with strong explanatory power are reserved while those with weak explanatory power are eliminated (*Chemura, Mutanga & Dube, 2017*).

## Set pair analysis

Set pair analysis integrates uncertainty analysis and determinacy analysis (*Zhao & Xuan, 1996*). The basis of SPA is set pair whose key is correlation degree, which has been applied as a criterion for the analysis of certainty and uncertainty (*Zhang et al., 2019*). During the hyperspectral band selection, the GC value $Y_i = (y_1, y_2, \cdots, y_n)$ can be defined as set A, and VIP value $Z_i = (z_1, z_2, \cdots, z_n)$ as set B, and then the two sets compose a set pair H = (A, B). To study the correlation degree of the set pair, the formulas are as follows:

$$\mu_i = S_i + F_i * I_i + P_i * J = \frac{s_i}{n} + \frac{f_i}{n} I_i + \frac{p_i}{n} J \tag{5}$$

$$I_i = \frac{S_i - P_i}{S_i + P_i} \tag{6}$$

$$v_i = 1/n + 1/n * u_i \tag{7}$$

$$W_i = \frac{v_i}{\sum_i^N v_i} (i = 1, 2, \ldots, N) \tag{8}$$

where $S_i$, $F_i$, $P_i$ are the identity degree, discrepancy degree and opposite degree of the two sets in the context of the same problem, respectively. They describe the association of the two sets from different aspects. $S_i$, $F_i$, $P_i$ meet the relationship: $S_i + F_i + P_i = 1$, $n$ denotes the total number of characteristics of the set pair; $s_i$ is the number of the common characteristics of the two sets; $f_i$ is the number of the different characteristics of the two sets (*Li et al., 2016*); $p_i$ is the number of the opposite characteristics of the two sets. When $J = -1$, the contact number $u_i$, the relative membership degree $v_i$, and the weighted value, $W_i$, of the VIP value and GC value, were calculated.

## Model construction and validation

### Classification of modeling and validation sets

Kennard-Stone algorithm (K-S) (*Kennard & Stone, 1969*) was used to classify the sample sets to select representative calibration sample sets. K-S was used to calculate the minimum Euclidean distance of the unselected samples according to the selected samples. Then the samples with the maximum Euclidean distance were selected into the calibration set in a repeated way until the samples of a specified number were selected. K-S algorithm has turned out to be effective in selecting representative samples (*Morais et al., 2019*).

Two-third of the water quality samples were selected for the modeling set ($n$ = 58) and one-third for the validation set ($n$ = 29).

### Modeling methods and evaluation

The PLSR and ELM methods were applied to the quantitative inversion of different water quality parameters in this paper. The PLSR model has been widely applied and can overcome the multicollinearity of independent variables because of its dimensionality reduction, information integration and band optimization in the modeling process, which has greatly improved the ability of the system to extract principal components (*Ryan & Ali, 2016*; *Wang et al., 2017b*). The ELM is a machine learning algorithm based on feed forward neural network. It has the advantages of high learning efficiency, high accuracy and simple parameter adjustment over the traditional feed forward neural network (*Song et al., 2013*). The optimal model was selected to inverse the water quality parameters through comparing the models' root mean square error of calibration (RMSEC), determination of coefficients ($R_c^2$), root mean square error of prediction (RMSEP), prediction determination of coefficients ($R_p^2$) and relative prediction deviation (RPD). $R_c^2$ and $R_p^2$ are used to evaluate the stability degree of a model. The closer its value is to 1, the higher stability the model has. RMSEC and RMSEP represent the accuracy of the model. The smaller their value is, the higher the model accuracy is. A model is considered excellent when RPD > 3.0, very good when 2.5 < RPD ≤ 3, good when 2 < RPD ≤ 2.5, satisfactory when 1.5 < RPD ≤ 2 and poor when RPD ≤ 1.5 (*Williams, 2001*).

## RESULT AND ANALYSIS

### Correlation between water quality parameters content and spectral reflectance

The Pearson correlation coefficients between each water quality parameter and the $R_{raw-SNV}$ (400–2,400 nm) were tested with the significance level ($P < 0.001$, $|r| = 0.324$ or above). The curves of correlation coefficients of water quality parameters were plotted in Fig. 2.

The change of the correlation coefficients of water quality parameters was complex. There were significant differences in the correlation coefficients between different water quality parameters and the wavelengths but their curve patterns were similar (*Peng et al., 2018*). There were many concentrated similar sensitive ranges but in general, the curves took "sharp fall, fluctuation, and oscillation" (Fig. 2). The COD, BOD, $NH_3$-N and TA were all highly correlated and represented a positive correlation (the correlation coefficients > 0.6) at the wavelengths of 900–1,300 nm and 1,500–1,800 nm. In contrast, the correlation between the TDS and spectral data and that between the TH and spectral data were not so satisfied (the correlation coefficients <0.6). The six parameters could pass the test of significance ($P = 0.001$) at the wavelengths of 400–450 nm and 850–1,800 nm. The band range was wide and the peak value of the correlation coefficients was relatively flat, so it was difficult to screen the characteristic bands based on the correlation coefficients.

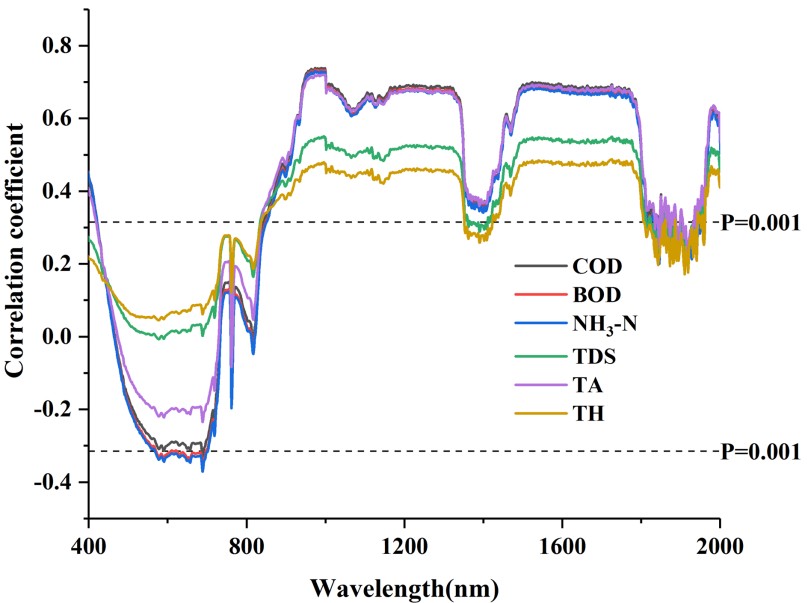

**Figure 2 Correlation coefficients of water quality parameters with standard normal variable reflectance.**

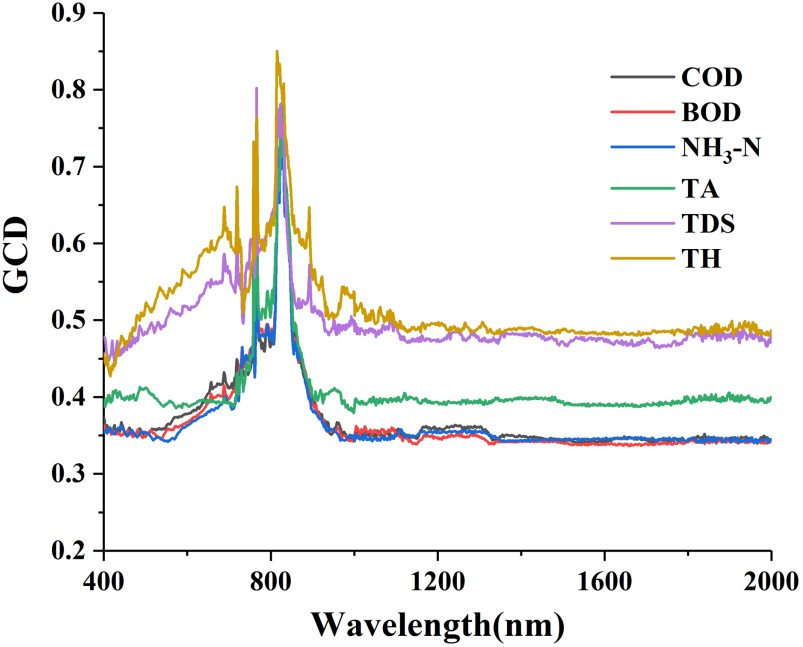

**Figure 3 Gray correlation degree (GCD) for water quality parameters with standard normal variable reflectance.**

## Selection of characteristic wavelength

### GC-based selection of characteristic wavelength

The curves of GCD for water quality parameters and $R_{raw-SNV}$ are shown in Fig. 3. An obvious peak wavelength appeared in the curves with the similar patterns of

**Table 2 Maximum gray correlation degree and band intervals of water quality parameters content with standard normal variable reflectance.**

| Water quality parameters | Sensitive band numbers | Maximum GCD | Maximum GCD band intervals (nm) |
|---|---|---|---|
| COD | 49 | 0.7949 | 820–830 |
| BOD | 50 | 0.7974 | 820–830 |
| NH$_3$-N | 46 | 0.7973 | 820–830 |
| TA | 93 | 0.7878 | 820–830 |
| TDS | 381 | 0.802 | 815–825/766 |
| TH | 601 | 0.8504 | 815–825 |

"sharp rise, fall and stabilization" on the whole (Fig. 3). The gray value of each water quality parameter gradually went up from 400 nm to a peak at about 820 nm, and then gradually down to 1,000 nm where it remained flat. The overall gray values of TDS and TH were relatively high, generally above 0.5, while those of COD, BOD, NH$_3$-N and TA were generally between 0.3 and 0.4 with only a few peak bands greater than 0.4 and above 0.8 at the most. It reflected that the GC method excels in eliminating a great amount of redundant information and selecting a few most sensitive bands.

The sensitive bands of the six parameters were counted based on their GCD analysis (Table 2). Of all the GCD values, the maximum ones were concentrated within the spectral wavelengths of 815–830 nm. The numbers of the sensitive bands were sequenced from large to small as: TH (601) > TDS (381) > TA (93) > BOD (50) > COD (49) > NH$_3$-N (46), and those of the maximum GCD values as: TH (0.8504) > TDS (0.802) > BOD (0.7974) > NH$_3$-N (0.7973) > COD (0.7949) > TA (0.7878). The comparison between the correlation coefficients and the GCD indicated a great difference: the higher correlation coefficient the parameters had, the lower the GCD values was, and the smaller number of sensitive bands were selected. This result revealed that for the parameters with strong spectral response the GCD method had better band screening ability.

### VIP-based selection of characteristic wavelength

The curves of VIP scores for water quality parameters and R$_{raw-SNV}$ are shown in Fig. 4. The curves patterns of the six parameters were similar and the overall scores of the VIP were relatively high (Fig. 4). These curves exhibited a sharp drop in the wavelength intervals of 400–450 nm, a sharp rise in 460–500 nm, a violent fluctuation in 700–1,000 nm, but in 1,100–2,000 nm TH and TDS displayed a less violent fluctuation and COD, BOD, NH$_3$-N and TA showed flat fluctuation. Many peak points existed in the VIP curves, and their peak intervals were relatively scattered, completely different from the single peak value in the GC curves. The principle of VIP > 1 was used to select and count the spectral sensitive bands of the six parameters (Table 3). The comparison of the maximum spectral response bands showed five parameters were found in the wavelength intervals of 460–480 nm except for NH$_3$-N in 990–999 nm.

The numbers of sensitive bands selected through VIP method were sequenced from large to small as BOD (770) > NH$_3$-N (768) > COD (753) > TA (709) > TH (543) >
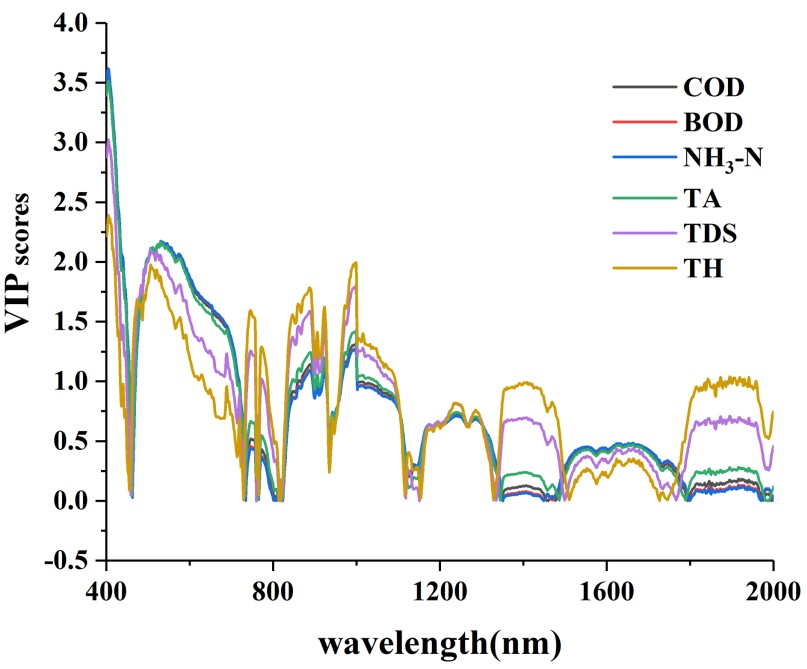

**Table 3 The maximum VIP scores of water quality parameters with standard normal variable reflectance.**

| Water quality parameters | Sensitive band numbers | Maximum VIP scores | Maximum VIP band interval (nm) | Maximum VIP band (nm) |
|---|---|---|---|---|
| COD | 753 | 1.634 | 465–475 | 466 |
| BOD | 770 | 1.439 | 464–474 | 762 |
| NH$_3$-N | 768 | 1.397 | 990–999 | 999 |
| TA | 709 | 2.466 | 460–470 | 464 |
| TDS | 497 | 4.275 | 460–470 | 463 |
| TH | 543 | 4.893 | 460–470 | 463 |

TDS (497), and those of the maximum VIP scores as: TH (4.893) > TDS (4.275) > TA (2.466) > COD (1.634) > BOD (1.439) > NH$_3$-N (1.397). The comparison between the two sequences indicated a great difference: the larger the VIP score was, the smaller number of selected sensitive bands was. On the whole, the effective bands could be retained as many as possible with the VIP method during bands screening.

### SPA-based selection of characteristic wavelength

The curves of SPA scores for water quality parameters and R$_{raw-SNV}$ are shown in Fig. 5. The curves patterns of TDS and TH were similar and exhibited a gentle flatness except for obvious peaks at the wavelengths of about 460 and 990 nm. Those of BOD, NH$_3$-N and TA were similar and showed notable valleys at the wavelengths of about 500 and 700 nm and notable peaks at the wavelengths of about 760 and 1,000 nm. Those of COD

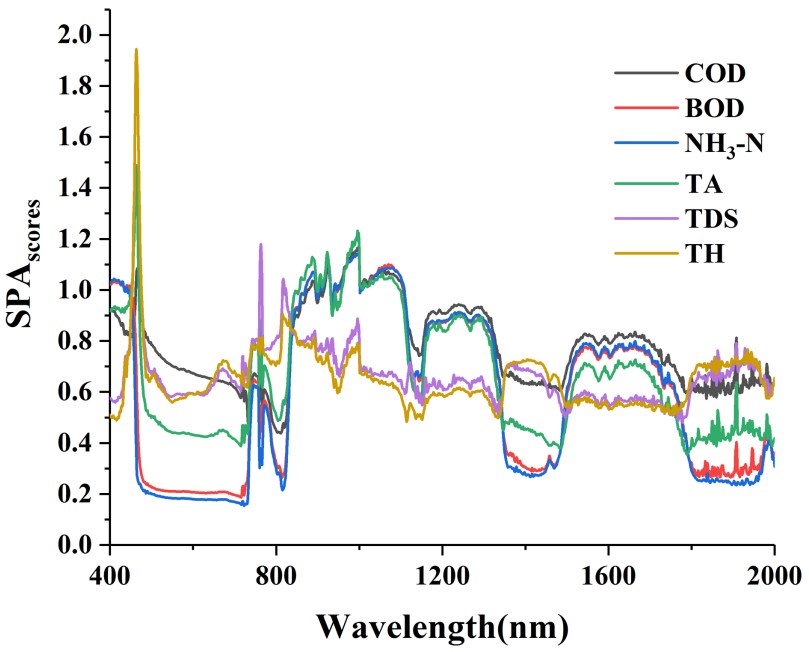

**Figure 5 The SPA scores of water quality parameters with standard normal variable reflectance.**

**Table 4 The Set pair analysis (SPA) scores of water quality parameter with standard normal variable reflectance.**

| Water quality parameters | Sensitive band numbers | Maximum SPA | Maximum SPA band interval (nm) | Maximum SPA band (nm) |
|---|---|---|---|---|
| COD | 750 | 1.598 | 465–475 | 466 |
| BOD | 767 | 1.409 | 762–763 | 762 |
| $NH_3$-N | 765 | 1.364 | 990–999 | 999 |
| TA | 696 | 2.409 | 460–470 | 464 |
| TDS | 280 | 1.928 | 460–470 | 463 |
| TH | 223 | 1.944 | 460–470 | 463 |

displayed notable peaks at the wavelengths of about 460 and 1,000 nm and fluctuation at other wavelengths. On the whole, great difference existed among the six curves, and the sensitive interval of each parameter was quite notable and different from each other.

The principles of GCD > 0.5 and VIP > 1 were used to select and count the spectral sensitive bands of the six parameters (Table 4). The comparison of the maximum spectral response bands revealed $NH_3$-N was in the wavelength intervals of 990–999 nm, BOD in 760–770 nm and the other four in 460–475nm. The numbers of sensitive bands selected through SPA method were arranged in a descending order as: BOD (767) > $NH_3$-N (765) > COD (750) > TA (696) > TDS (280) > TH (223), the scores of the maximum SPA as: TA (2.409) >TH (1.944) > TDS (1.928) > COD (1.598) > BOD (1.409) > $NH_3$-N (1.364). The comparison between the two orders indicated a great difference: the larger the SPA score was, the smaller number of the selected sensitive bands had.

**Table 5 PLSR model results based on full band, GC, SPA and VIP screening band.**

| Water quality parameters | Wavelength selection methods | Main factor numbers | Modeling set $R_c^2$ | Validation set $R_p^2$ | Relative percent deviation RPD | Robustness Robust |
|---|---|---|---|---|---|---|
| COD | GC | 6 | 0.970 | 0.954 | 4.690 | 0.984 |
| | VIP | 4 | 0.938 | 0.917 | 3.238 | 0.978 |
| | SPA | 4 | 0.938 | 0.917 | 3.237 | 0.978 |
| | All | 9 | 0.992 | 0.956 | 4.532 | 0.964 |
| BOD | GC | 5 | 0.976 | 0.965 | 5.335 | 0.988 |
| | VIP | 10 | 0.992 | 0.951 | 4.424 | 0.958 |
| | SPA | 4 | 0.958 | 0.930 | 3.348 | 0.971 |
| | All | 7 | 0.990 | 0.960 | 4.998 | 0.970 |
| NH₃-N | GC | 5 | 0.972 | 0.970 | 5.894 | 0.998 |
| | VIP | 7 | 0.970 | 0.935 | 3.977 | 0.965 |
| | SPA | 7 | 0.969 | 0.935 | 3.974 | 0.965 |
| | All | 8 | 0.985 | 0.962 | 5.228 | 0.977 |
| TDS | GC | 8 | 0.876 | 0.772 | 2.049 | 0.880 |
| | VIP | 3 | 0.767 | 0.758 | 1.892 | 0.988 |
| | SPA | 5 | 0.820 | 0.793 | 2.150 | 0.964 |
| | All | 5 | 0.862 | 0.791 | 2.126 | 0.918 |
| TA | GC | 5 | 0.917 | 0.890 | 2.963 | 0.971 |
| | VIP | 4 | 0.883 | 0.927 | 3.578 | 1.049 |
| | SPA | 4 | 0.883 | 0.927 | 3.588 | 1.050 |
| | All | 7 | 0.956 | 0.921 | 3.206 | 0.963 |
| TH | GC | 12 | 0.978 | 0.900 | 2.871 | 0.920 |
| | VIP | 4 | 0.779 | 0.733 | 1.764 | 0.941 |
| | SPA | 3 | 0.794 | 0.708 | 1.701 | 0.892 |
| | All | 5 | 0.820 | 0.817 | 2.228 | 0.996 |

## Construction and analysis of PLSR model

The sensitive bands selected by different methods as GC, VIP and SPA were applied to PLSR modeling. The results of PLSR model are shown in Table 5.

In general, the PLSR model had a good spectral prediction of water quality parameters (*Wang et al., 2019*). The GC-PLSR model was proved to be optimal for COD, BOD, NH3-N and TH with the best prediction. SPA-PLSR model was proved to be optimal for TDS and TA with the best prediction (Table 5). The VIP-PLSR model of BOD had the best modeling effect ($R_c^2 = 0.99$). The GC-PLSR model of NH₃-N had the best prediction effect ($R_p^2 = 0.962$, RPD = 5.894). The inversion effects of TDS and TH were relatively less satisfying, with the $R_c^2$ and $R_p^2$ generally around 0.8. Overall, the PLSR models based on the characteristic bands of the six parameters exhibited good modeling and prediction effect as well as a greater improvement than those full band-based models.

## Construction and analysis of ELM model

The sensitive bands selected using GC, VIP and SPA methods were applied to build ELM model. The results of ELM model are shown in Table 6.

**Table 6 ELM model results based on full band, GC, SPA and VIP screening band.**

| Water quality parameters | Wavelength selection methods | The number of neurons in the hidden layer | Modeling set $R^2c$ | Validation set $R^2p$ | Relative percent deviation RPD | Robustness Robust |
|---|---|---|---|---|---|---|
| COD | GC | 23 | 0.964 | 0.956 | 4.667 | 0.991 |
| | VIP | 51 | 0.884 | 0.885 | 2.892 | 1.001 |
| | SPA | 55 | 0.959 | 0.928 | 3.567 | 0.967 |
| | All | 48 | 0.936 | 0.920 | 3.513 | 0.983 |
| BOD | GC | 230 | 0.986 | 0.976 | 6.192 | 0.989 |
| | VIP | 36 | 0.949 | 0.933 | 3.872 | 0.983 |
| | SPA | 37 | 0.939 | 0.930 | 3.783 | 0.990 |
| | All | 30 | 0.914 | 0.876 | 2.764 | 0.959 |
| NH$_3$-N | GC | 200 | 0.979 | 0.976 | 6.596 | 0.997 |
| | VIP | 144 | 0.982 | 0.965 | 5.391 | 0.983 |
| | SPA | 37 | 0.960 | 0.959 | 4.901 | 1.019 |
| | All | 48 | 0.949 | 0.940 | 4.155 | 1.093 |
| TDS | GC | 23 | 0.595 | 0.613 | 1.431 | 1.029 |
| | VIP | 19 | 0.714 | 0.715 | 1.822 | 1.001 |
| | SPA | 50 | 0.820 | 0.790 | 2.198 | 0.964 |
| | All | 48 | 0.735 | 0.706 | 1.627 | 0.961 |
| TA | GC | 103 | 0.907 | 0.895 | 3.059 | 0.986 |
| | VIP | 22 | 0.827 | 0.828 | 2.150 | 1.001 |
| | SPA | 68 | 0.952 | 0.924 | 3.651 | 0.970 |
| | All | 30 | 0.778 | 0.784 | 2.089 | 1.008 |
| TH | GC | 6 | 0.570 | 0.560 | 1.250 | 0.982 |
| | VIP | 45 | 0.673 | 0.688 | 1.606 | 1.022 |
| | SPA | 37 | 0.937 | 0.910 | 3.358 | 0.971 |
| | All | 32 | 0.535 | 0.497 | 1.132 | 0.930 |

The ELM models based on the characteristic bands of the six parameters exhibited good modeling and prediction effect as well as a greater improvement than those based on the full band. The GC-ELM model was proved optimal for COD, BOD and NH$_3$-N with the best prediction. SPA-ELM was proved optimal for TDS, TA and TH with the best prediction. The VIP-ELM model of BOD had the best modeling effect ($R^2_c = 0.986$). The GC-ELM model of NH$_3$-N exhibited the best prediction effect ($R^2_P = 0.976$, RPD = 6.596). The ELM models of COD, TH and TA had the satisfying modeling and validation effect. The validation effect of TDS was relatively less satisfying ($R^2_c = 0.820$, $R^2_P = 0.790$, RPD = 2.198).

# DISCUSSION

## Comparison among the estimating results of different water quality parameters

The optimal wavelength selection methods varied when the optimal modeling methods were different (Tables 5 and 6). The hyper-spectrally estimated value and the chemically

measured value of the six water quality parameters were compared under the optimal model (Fig. 6).

Different water components can change the spectral radiation by their own absorption and scattering characteristics of light. The water components (such as SPM, CDOM, phytoplankton, blue-green algae) can be obtained by separating the factors of various parameters from the total water radiation (*Dörnhöfer & Oppelt, 2016*). However, the problem of "lower ion content and weaker spectral response" often exists in the spectral analysis of water components, and was also reflected in the verification results of this study. All the six parameters produced satisfying inversion results, which indicated the feasibility of quantitative inversion of sewage water quality parameters by hyperspectrum (*Pu et al., 2017*). The sequence of the predicting power of the water quality parameters was $NH_3$-N > BOD > COD > TA > TH > TDS. The validation results showed that most data points of the five water parameters, $NH_3$-N, BOD, COD, TA and TH, were concentrated near line 1:1. This indicated that the hyperspectral analysis values of the five parameters were very close to the chemically estimated values and the optimal models of these five parameters had strong prediction ability (RPD > 3.0) (Tables 5 and 6). Comparatively, due to its lower content, the data points of TDS were relatively discrete, indicating that the model prediction ability was average (RPD = 2.198).

## Correlation analysis and inversion performance

The raw spectral reflectance curve of each water sample exhibited different shapes (Fig. 2A). This difference results from the inconsistency between the contents and types of the water quality parameters, and the different characteristics of spectrum obtained through absorption and scattering of light. The results accord with those in previous studies (*Abdelmalik, 2018*; *Abd-Elrahman et al., 2011*; *Rostom et al., 2017*; *Wang, Pu & Sun, 2016*), which demonstrates that the VIS-NIR spectrum can be used to quantitatively determine water quality parameters.

Traditionally, correlation analysis is helpful to reveal the relationship between water quality parameters and the VIS-NIR spectrum (*Peng et al., 2018*). In this paper, the numbers of the significant wavebands of the six water quality parameters can be arranged in an order, in which the numbers of $NH_3$-N, BOD and COD were close to one another but larger than those of the other three parameters. The number of TA followed, and those of TH and TDS came last. This order agreed with that of their correlation coefficient ranges. Therefore, significant correlations exist between $NH_3$-N, BOD, COD and their reflectance spectrum, which reflects the optimal models of the three parameters have a very good prediction performance (Fig. 6). By contrast, the correlations between the other three water quality parameters and their reflectance spectrum were relatively low, which led to poor model prediction performance, especially for the TDS model. However, by comparing the three screening methods, the numbers of the sensitive wavebands of TA, TH and TDS were not as small as had been expected (Tables 3, 4 and 5). This is because different waveband screening methods use different calculation mechanisms, which leads to different methods to screen optimal response spectrum for different parameters (*Zhang, Maoguo & Yongqiang, 2018*). From the perspective of

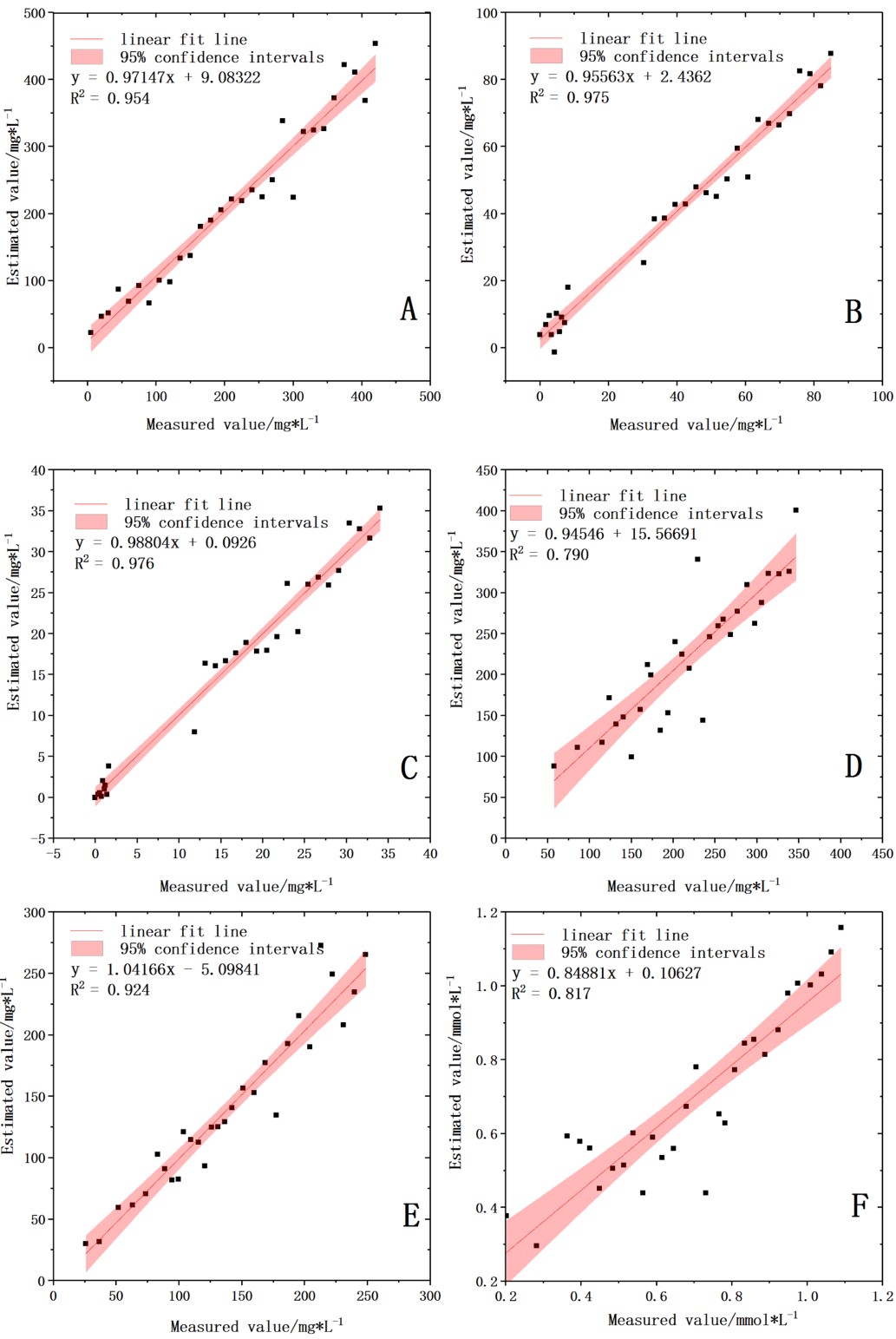

**Figure 6 Validation of water quality parameters based on the best model.** (A) COD with GC-PLSR model. (B) BOD with GC-ELM model. (C) NH₃-N with GC-ELM model. (D) TDS with SPA-ELM model. (E) TA with SPA-ELM model. (F) TH with SPA-ELM model.

modeling methods, ELM model performed better than PLSR model, which also indicated the superiority and strong learning ability of machine learning algorithm (*Keller et al., 2018*). From the perspective of band selection methods, GC and SPA are superior to VIP. Therefore, the selection of band screening methods will be of practical significance for the quantitative inversion of water quality parameters in the future research.

## Effects of wavelength selection on hyperspectral estimation models

In the process of hyperspectral analysis, due to the mass of hyperspectral information, there exists a large amount of redundant information irrelevant to parameters content. This makes it difficult to mine and extract effective information. This problem complicates the hyperspectral analysis and decrease the analysis accuracy, which hinders the development and utilization of related testing equipment. Therefore, characteristic selection of hyperspectral data information is required. The characteristic selection methods are devised to find the smallest subset from the original characteristics without reducing the accuracy (*Zhang, Maoguo & Yongqiang, 2018*). For the hyperspectral band selection, there are two key aspects, namely, effective information preservation and redundancy elimination, which has a great impact on subsequent modeling and validation.

In this study, GC and VIP methods were used to explore their spectral characteristic response intervals for six main water quality parameters in wastewater. These two methods had their fixed preferences in terms of information preservation and redundancy elimination. Therefore, the two methods had quite different performance in the band selection for the parameters and modeling methods, and their performance was very unstable (*Wang et al., 2019*). Of the two methods, the GC method had better performance in redundancy elimination. Better spectral response of the water quality parameters means smaller number of bands filtered by the GC method, so the GC method is the best screening method for COD, BOD and $NH_3$-N with a satisfying screening performance in this study.

However, the VIP method performed better in information preservation, and it could retain as many spectral band intervals with good response to the inversion parameters as possible (*Oussama et al., 2012*; *De Almeida et al., 2013*). So, the VIP method retained the largest number of bands during band selection. Meanwhile, compared with the GC method, VIP was more applicable to characteristic band selection of TA and TDS.

As to this condition, some researchers considered it necessary to make the optimal trade-offs between information preservation and redundancy elimination according to the different characteristics of the datasets, and have conducted some explorations (*Zhang, Maoguo & Yongqiang, 2018*). This study put forward a band screening method based on SPA to optimize the trade-offs between GC and VIP. From its application to the datasets in this study, the SPA method could well balance effective information retaining and redundant information eliminating. Compared with GC and VIP, the SPA method had a

more stable ability of band screening. It had a strong applicability to predicting the six water quality parameters as well as building the linear modeling method PLSR and the nonlinear modeling method ELM. Therefore, the SPA method had the superiority in certainty and uncertainty analysis of characteristic bands, and it is applicable to the integrative optimization of GC and VIP.

## Research limitations and future research

This study was only the statistical application of characteristic band screening methods to the inversion of different water quality parameters and failed to explore the spectral response mechanism of the various water quality parameters (*Hu & Wang, 2017*), which would lead to some limitations in the applicability of the model. Meanwhile, the optimized band screening method based on the SPA needs further study, which can focus on further:

1. Exploring the different spectral sensitive intervals and analyzing the spectral response mechanism of each water quality parameter.
2. Probing into the band screening mechanism and optimizing various screening methods so as to propose a band screening method suitable for various models; and applying various screening methods at the same time in the longitudinal study to achieve multiple screening.

## CONCLUSIONS

This paper studied the feasibility of quantitatively estimating water quality parameters via VIS-NIR spectral model. The GC, VIP and SPA methods were used to screen the sensitive intervals based on the spectral responses of different water quality parameters in sewage. The inversion accuracy of the six water quality parameters by the linear PLSR and nonlinear ELM modeling methods was compared, and finally the research arrived at the following conclusions:

1. Band screening plays an important role in spectral data processing and spectral model accuracy improvement. In this study, the GC had the best performance in redundancy elimination (up to 97%), the VIP performed best in information preservation, and the SPA could make the optimal trade-offs between information preservation and redundancy reduction, so the SPA had the best applicability.
2. Both the PLSR and ELM models have good performance in modeling and validation. The PLSR model was more applicable and the ELM model had higher prediction accuracy. Among the six corresponding optimal models for the six water quality parameters, five were ELM models while only one was a PLSR model.
3. The optimal spectral inversion models of the six water quality parameters had quite different validation results: the prediction of the COD, BOD, $NH_3$-N and TA models was quite satisfying (their RPD values are 4.69, 6.192, 6.596 and 3.651, respectively), while that of the TH and TDS was relatively poor.

## ACKNOWLEDGEMENTS

The authors want to thank A.P. Ying Meng and Yinwen Chen for their help in language standardization of this manuscript and providing helpful suggestions. We are especially grateful to the reviewers and editors for appraising our manuscript and for offering instructive comments.

### Funding

This research is supported by National Natural Science Foundation of China (51409221, 51979234), National Key Research and Development Program of China (2017YFC0403302) and Humanities and Social Science Program of Northwest A&F University (Z109021405). The funders had no role in study design, data collection and analysis, decision to publish, or preparation of the manuscript.

### Grant Disclosures

The following grant information was disclosed by the authors:
National Natural Science Foundation of China: 51409221, 51979234.
National Key Research and Development Program of China: 2017YFC0403302.
Humanities and Social Science Program of Northwest A&F University: Z109021405.

### Competing Interests

The authors declare that they have no competing interests.

### Author Contributions

- Zheng Xing conceived and designed the experiments, performed the experiments, analyzed the data, contributed reagents/materials/analysis tools, prepared figures and/or tables, authored or reviewed drafts of the paper, approved the final draft.
- Junying Chen conceived and designed the experiments, authored or reviewed drafts of the paper, approved the final draft.
- Xiao Zhao conceived and designed the experiments, authored or reviewed drafts of the paper, approved the final draft.
- Yu Li conceived and designed the experiments, contributed reagents/materials/analysis tools, authored or reviewed drafts of the paper, approved the final draft.
- Xianwen Li conceived and designed the experiments, authored or reviewed drafts of the paper, approved the final draft.
- Zhitao Zhang conceived and designed the experiments, contributed reagents/materials/analysis tools, authored or reviewed drafts of the paper, approved the final draft.
- Congcong Lao conceived and designed the experiments, prepared figures and/or tables, approved the final draft.
- Haifeng Wang conceived and designed the experiments, prepared figures and/or tables, approved the final draft.

## Field Study Permissions

The following information was supplied relating to field study approvals (i.e., approving body and any reference numbers):

Field experiments were approved by the domestic sewage treatment plant, Zhouzhi, Shaanxi, China (No. 51409221, 51979234).

## Data Availability

The raw measurements are available as a Supplemental File.

## Supplemental Information

Supplemental information for this article can be found online at http://dx.doi.org/10.7717/peerj.8255#supplemental-information.

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
