# Peer review of "Quantitative estimation of wastewater quality parameters by hyperspectral band screening using GC, VIP and SPA"

_PeerJ, doi:10.7717/peerj.8255_

## Round 0.1 · original submission · Minor Revisions

The reviewers recommended minor to major revisions. Based on my consideration of their reviews, I am recommending minor revisions. Two reviewers mentioned the need for improved presentation. Therefore, in your revision please give significant attention to language, grammar and careful use of correct terminology. I would recommend that you have the manuscript reviewed by a technical writer that is fluent in English for this purpose. The two primary technical concerns expressed were 1) a perceived lack of comparison between directly measured water quality variables and hyperspectral estimates (Reviewer #1); and 2) a perceived lack of validation against data not used in the calibration (Reviewer #3). In my own reading of the manuscript, I believe that both of these concerns reflect inadequate understanding of the manuscript rather than an actual technical deficiency. You do actually show comparisons between direct and indirect measurements, and as I understand you held back some of the data from the training set for use in validation (as is common in machine learning methods). Therefore I think that you only need to make sure that these points are more clear to the readers. Please modify the manuscript as needed to make these issues more clear (perhaps discussing these two points in the introduction or conclusion rather than just in the technical description of the methods). In your response, please provide a detailed description of how you have addressed each of the reviewers' comments.

Reviewer 1 ·

Basic reporting

1. Manuscript is lacking a good English. The language should be improved to ensure that it can be in publishable form and clear to reader. Some examples where the language could be improved include lines 59, 63, 81, 97, 101, 93, 94, 109, 116- the current phrasing makes comprehension unclear and difficult.
Line 81: it should be “water sampling at fixed points” for “in fixed”, "physical and chemical examinations" for “experiments”
Line 97: Authors have written “particles” for COD and BOD “main pollutant particles (COD, BOD)”. Please describe is it “parameters”? or How these can be particles?
Line 101: Hyper-spectral techniques are still being widely used for various monitoring of natural resources. This line needs to be corrected.
Line 160: Seems incomplete and contains spelling mistake.
Line 169: Needs to be revised and corrected with “water taking” for "water taken", "were shown" should be replaced by “are shown”.
There is no space between the word and bracket (eg. substances(TDS)), that needs to be corrected.
2. Literature references are not sufficient and have not been focused on latest findings.
3. Figure and table legends need to be revised and explained. Units should be in standard format and representation of data corrected (eg “90%-98%” should be as “90-98%”).

Experimental design

Material and methods sections needs some latest references. What was the purpose of adding pure water to the wastewater samples before the analysis, please explain?
Please explain “what is the Table 1 for?” Is it chemically estimated values of the wastewater? This should be properly mentioned in the material method section.

Validity of the findings

Result and discussions is lacking sufficient and proper references in support of the findings, that needs to be addressed in the MS.
Compared representation of chemically estimated and hyper-spectral analysis values is lacking in MS and this needs to be addressed.

Additional comments

Figure and table legends needs to be revised and explained. Units should be in standard format and representation of data corrected (eg “90%-98%” should be as “90-98%”). Axis title “Wavelength/nm” is confusing, please explain what it is for or correct it as “Wavelength (nm)”. Other axis titles also need correction. Following the published paper of the journal will be helpful in this regard.

Annotated reviews are not available for download in order to protect the identity of reviewers who chose to remain anonymous.

Reviewer 2 ·

Basic reporting

The language and grammar of the manuscript needs to be improved. The references, figures, tables are alright.

Experimental design

The manuscript deals with original research and falls within the aim and scope of the journal. The research question is well defined, meaningful, and relevant. Methodology has been described that could be replicated.

Validity of the findings

The data provided are robust, statistically sound and controlled. Conclusions are well stated and supports the result.

Additional comments

Line # 97: In the water, some of the main pollutant particles (COD, BOD)
Comment: COD and BOD are not pollutant particles. Instead, they are pollution indicators.
Line # 175-176: The laboratory was kept in constant temperature and protected during the experiments.
Comment: What does the statement ‘laboratory was kept in constant temperature...’ refers to.

·

Basic reporting

Environmental pollution is one of the most challenging problem against mankind. The measurement of water quality parameter regularly is a challenging job. The regular measurement is costly affair. Keeping this fact in mind. the present work is quite promising.

Experimental design

Experimental design part is ok.

Validity of the findings

The model results showing good agreement with experimental data but the source of data and data used for model development is from same source. It is not clear that the model will work for any other set of sample taken from different source. Rigorous validation using waste water from different sources is required to establish the efficacy and applicability of this method.

Additional comments

The authors should include validation of best model taking waste water from different sources.

---

## Round 0.2 · Minor Revisions

Thank you for the revised manuscript and for your efforts addressing the reviewers' comments. In my opinion, you have adequately addressed the comments with one exception: A clearer explanation of why the samples were diluted with "pure water" prior to hyperspectral analysis is still needed (as suggested initially by Reviewer #1).

In lines 170-174 of the revised manuscript you state that: "The difference of the quality of these water samples was too big to fully represent varied levels of wastewater quality. To enhance the distribution gradient and the representativeness of samples, we added varied amount of pure water to the wastewater samples from different sources before the analysis. Eighty-seven water samples with a COD 5 mg/L gradient unit of experiment were obtained."

This statement is still quite confusing in my opinion. I understand that you took samples from different locations within the treatment plan (lines 164-166), and that variation in water quality among the samples was large. However, stating that the difference was too big to represent varied levels doesn't make any sense to me. In fact, a large range of water quality does represent varied levels quite well. Perhaps what you mean is that the variation was too large to be measured by the hyperspectral method? And so therefore you diluted some of the most polluted samples to ensure they were within the range of sensitivity of the method? What is a "distribution gradient" and "5 mg/L gradient unit of experiment"? This terminology also makes no sense to me and is not defined here as far as I can tell. As long as the degree of dilution was tracked for each sample, I don't see any problem with diluting the samples if there is a good reason. However, the statement on lines 170-174 does not provide an understandable explanation.

Also, you had already performed laboratory measurements on these samples, so you knew how much they should be diluted. In a real application (in which you want to use hyperspectral instead of laboratory measurements) how would you know what level of dilution would be appropriate?

The remainder of the revisions are good and have improved the quality of the manuscript significantly. Please provide the modifications necessary to clarify the above confusions (shared by Reviewer #1) and if appropriately addressed I believe the manuscript will be ready for acceptance.

---

## Round 0.3 · accepted · Accept

Thank you for your most recent revisions. This explanation is much clearer and will avoid confusion by the reader as to your reasons for diluting some of the samples.